# Neutralizing Antibody Response to Genotypically Diverse Measles Viruses in Clinically Suspected Measles Cases

**DOI:** 10.3390/v15112243

**Published:** 2023-11-10

**Authors:** Sunil R. Vaidya, Neelakshi S. Kumbhar, Gargi K. Andhare, Nilesh Pawar, Atul M. Walimbe, Meenal Kinikar, Sunitha M. Kasibhatla, Urmila Kulkarni-Kale

**Affiliations:** 1ICMR-National Institute of Virology, 20-A, Dr. Ambedkar Road, Pune 411001, India; 2Bioinformatics Centre, Savitribai Phule Pune University, Pune 411007, India; meenalkinikar@gmail.com (M.K.);; 3HPC-Medical and Bioinformatics Applications Group, Centre for Development of Advanced Computing, Panchavati, Pashan, Pune 411008, India

**Keywords:** India, measles virus genotypes, neutralization activity, serological investigations, suspected measles cases

## Abstract

The neutralizing antibody (Nt-Ab) response to vaccine and wild-type measles viruses (MeV) was studied in suspected measles cases reported during the years 2012–2016. The neutralization activity against MeV A, D4 and D8 genotypes was studied on sera (Panel A; n = 68 (measles-immunized) and Panel B; n = 50 (unvaccinated)) that were either laboratory confirmed or not confirmed by the presence of IgM antibodies. Additionally, the Nt-Ab response in Panel A was measured against the MeV vaccine and four wild-type viruses. Neutralization results were compared using homology modeling and molecular dynamics simulation (MDS) of MeV-hemagglutinin (H) and fusion (F) proteins. Overall, the Nt-Ab titres for MeV-A were found to be significantly lower than MeV-D4 and MeV-D8 viruses for Panel A. No major difference was noted in Nt-Ab titres between MeV-D8 viruses (Jamnagar and New Delhi), whereas MeV-D4 (Sindhudurg and Bagalkot (BGK) viruses) showed significant differences between Nt-Ab titres for Panel B. Interestingly, the substitutions observed in epitopes of H-protein, L249P and G316A are observed to be unique to MeV-BGK. MDS of H-protein revealed significant fluctuations in neutralizing epitopes due to L249P substitution. The majority of the clinically suspected cases showed Nt-Abs to MeV wild-types. Higher IgG antibody avidity and Nt-Ab titres were noted in IgM-negatives than in IgM-positives cases, indicating reinfection or breakthrough. MDS revealed reduced neutralization due to decreased conformational flexibility in the H-epitope.

## 1. Introduction

Measles is a highly infectious disease caused by the measles virus (MeV), belonging to the genus *Morbillivirus* of family *Paramyxoviridae*. MeV is a negative stranded RNA virus, with a genome of 15,894 nucleotides that contains six structural genes: nucleocapsid [N], phosphoprotein [P], matrix [M], fusion [F], hemagglutinin [H] and large protein [L] and two non-structural [V and C] genes [1]. MeV causes high-grade fever, cough, coryza, conjunctivitis and maculopapular skin rashes. Koplik’s spots appear on buccal mucosa 1–2 days prior to skin rash and MeV spreads to various organs with serious CNS complications like ADEM, MIBE and SSPE [1]. Measles in vaccinated individuals could be due to either primary or secondary vaccine failures. Primary vaccine failure refers to vaccine recipients who failed to develop protective immunity after vaccination, which is possibly due to age at the time of vaccine administration, whereas secondary vaccine failure refers to those individuals who develop a protective immune response after vaccination but lose this protection over time [2]. Vaccine failure could also be due to inappropriate storage and handling, leading to decreased vaccine effectiveness [3]. The recent resurgence of measles among fully or partially immunized individuals raises concerns about the durability of vaccine induced immunity in the population [4].

The government of India introduced a two-dose strategy for measles vaccine administration (i.e., inclusion of the first dose in year 1985 and the second dose in year 2010) and combined measles–rubella vaccine administration in 2017 through the Universal Immunization Program [5,6,7]. As per the routine immunization schedule, every child should be given two doses of measles vaccine, the first at the age of 9–12 months and the second at the age of 16–24 months [5]. Detection of MeV-specific IgM antibodies using enzyme-linked immunosorbent assay (ELISA) is the most widely used method to confirm measles infection. Individuals with confirmed measles and a prior immunologic response to MeV (reinfection) from either vaccination or natural disease that occurred at least 4 months before the onset of symptoms, can be identified by the presence of a high-avidity measles IgG antibodies [8]. A person with high-avidity measles IgG antibodies can be considered as a laboratory confirmed measles case, along with those cases where there is IgM-antibody detection in serum and MeV-RNA detection in a throat swab or a urine specimen. High concentrations of measles neutralizing antibodies (Nt-Ab) have been observed among confirmed measles cases with high-avidity IgG antibodies [9]. However, ELISAs do not differentiate between neutralizing and non-neutralizing antibodies. Thus, for the characterization of vaccine or wild-type-induced immune response, rapid and reliable neutralization tests are essential [10]. The neutralization test measures antibodies that have the biological ability to neutralize MeV in vitro and is used for testing protective immunity.

A study from Iran revealed lower levels of Nt-Ab titres to MeV B3 genotype compared to the MeV genotypes A, D4 and H1 in individuals who were administered two doses of measles vaccine [11]. Sera from a group of healthcare workers from the UK showed that the titres for five circulating MeV genotypes (B3, C2, D4, D8 and H1) were significantly higher than the vaccine strain [12], whereas Japanese investigators demonstrated similar Nt-Ab titres for vaccine strain and circulating genotypes D3 and D5 strains in the serum samples obtained from the vaccine recipients [13]. Antigenic characterization of MeV wild-type (Chicago-1 strain, genotype D3) showed about five times higher titre compared to the vaccine strain [14].

MeV produces typical symptoms of high-grade fever, including maculopapular skin rashes, cough, coryza and conjunctivitis in humans, and infection caused by one MeV genotype may protect against subsequent infection by other MeV genotypes or prior immunization using a measles antigen (live attenuated vaccine). Interestingly, genome sequencing studies on Indian MeV wild-types revealed mutations in H-protein at important epitope sites, i.e., HNE, SSE, LE and NE for a few D4 and D8 isolates [15] that prompted us to test the hypothesis of the monotypic nature of MeVs using cross-neutralization and bioinformatics studies.

Thus, the present study was undertaken to understand the in vitro neutralization activity to MeV vaccine (genotype A) and the wild-type (genotype D4 with and without mutations in neutralization epitopes and genotype D8 with and without mutations in neutralization epitopes) [15], using two different serum panels, i.e., A (n = 68) and B (n = 50) that were referred for the laboratory diagnosis of measles. Of these panels, Panel A serum samples had a history of measles immunization (at least one dose) and were neutralized with vaccine and wild-type viruses in neutralization tests and Panel B serum samples were unvaccinated but suspected measles cases that were neutralized with both normal and mutated MeVs (D4 and D8 strains). In addition, the molecular basis for differential neutralization was investigated using homology modeling and molecular dynamics simulations. This study is useful in the context of the measles elimination program undertaken by the Government of India and other countries.

## 2. Materials and Methods

### 2.1. Serum Samples

Serum samples were obtained from 57 females and 61 males during years 2012–2016 who presented with fever with skin rashes along with either cough or coryza or conjunctivitis. Unfortunately, urine specimens/throats swabs were not available for MeV genotyping studies. The history of prior infection (before present episode) of measles/rubella if any was either not available or not collected from these cases. In these cases, the onset of fever with skin rashes was found to vary between 2 and 35 days. Of these 118 suspected cases, 21 were <1 year old, 71 were between 1 and 6 years old and remaining 26 were between 7 and 15 years old. All these samples were referred for measles diagnosis from different districts of Maharashtra state, of which 68 suspected cases (Panel A) had documented history of measles vaccine. However, information pertaining to the number of doses was not available. The Panel B of 50 suspected cases was unvaccinated for measles but reported clinical measles. These panels contained laboratory confirmed measles cases and cases for which acute MeV infection was not confirmed. Between years 2012 and 2016, occurrence of MeV genotypes D4 and D8 were reported in Maharashtra, India.

### 2.2. Measles Virus Specific IgM, IgG Antibody Detection

Measles virus-specific IgM and IgG antibodies in a panel of sera were measured using the commercial ELISA (NovaTec Immundiagnostica GmbH, Dietzenbach, Germany). Results of the samples were interpreted according to the criteria provided by the manufacturer. Samples with equivocal results were tested twice in different runs prior to confirmation. To understand the acute and past infections, all samples were tested in a MeV IgG antibody avidity ELISA (Abcam Inc., Cambridge, MA, USA) and results were interpreted as per kit criteria.

### 2.3. Viruses Used in Neutralization Tests

For the neutralization tests, serum samples were heat inactivated in a 56 °C water bath for 30 min and subsequently serum dilutions, i.e., 1:2 to 1:256 were prepared. Virus-serum neutralization reactions were performed by adding an equal volume of diluted virus at 37 °C in a 5% CO_2_ incubator for 2 h. For the neutralization experiments, MeV Edmonston Zagreb vaccine (Serum Institute of India Private Limited Pune, India), MeV D4 (Sindhudurg-2012_MH356255 and Bagalkot-2012_MH356248) and MeV D8 (Jamnagar-2016_MN131118 and New Delhi-2014_MN125026) viruses were diluted giving 35–45 viral foci per 10 µL inoculum per well and the same was used in the focus reduction neutralization test (FRNT). The non-neutralized MeV was detected using immunocolorimetric staining [10] (Figure 1) and 50% Nt-Ab titres were deduced using the Kärber formula [16]. MeV Nt-Ab titre ≥ 1:8 is considered as positive.

### 2.4. Homology Modeling of Hemagglutinin of Measles Virus Isolates

A 3D structure of hemagglutinin protein from Bagalkot isolates of MeV (genotype D4) was built using homology modeling approach. The structure of the Edmonston B strain was used as a template (A genotype; PDB ID: 2ZB6; Resolution: 2.60 Å [17]) and the regions missing in the template were built using low resolution structure of the same protein (PDB ID: 3ALZ; Resolutions: 4.15 Å [18]). The Modeller program [19] implemented in the Discovery Studio Toolkit (https://www.3ds.com/products-services/biovia/products/molecular-modeling-simulation/biovia-discovery-studio/, accessed on 22 October 2023) was used for building models. The initial structure was built by copying the coordinates of structurally conserved regions from the templates and the missing regions were modeled as loops using the Modeller program. The model was optimized using distance-dependent dielectric constant, CHARMM22 force field [20,21] and steepest descent method implemented in Discovery Studio. The stereochemistry and geometry of the predicted structure was evaluated using programs ProCheck [22] and ProSA [23,24].

### 2.5. Molecular Dynamics Simulation and Analysis

The MD simulations of both the Bagalkot isolate (harboring unique mutations L249P and G316A) and the Edmonston B strain (which was used as a reference), were carried out using GROMACS version 2020.1 [25]. The missing regions in the reference strain (PDB: 2ZB6) were modeled as loop regions prior to molecular dynamics (MD) simulations.

The parameters for MD simulations were derived from the CHARMM36 force field [26]. SPC/E model was used to represent water and ions were added to neutralize the system. Energy minimization was carried out using steepest descent. Equilibration was carried out for 2 ns consisting of 1 ns NVT followed by 1 ns NPT simulations. Temperature was kept constant at 300 K using a modified Berendsen thermostat [27]. Three independent production runs of 100 ns each were performed for the Bagalkot isolate (homology model) and Edmonston B (reference strain). MD trajectories were analyzed and root mean square deviation (RMSD) was calculated for the whole protein as well as for the respective epitope regions (neutralizing epitope (NE) and loop epitope (LE) regions). Root mean square fluctuation (RMSF) was calculated over the period of simulation. The conformational fluctuations in NE and LE between MeV Bagalkot isolate and Edmonston B strain were analyzed.

### 2.6. Statistical Analysis

The GMTs were compared by using *t*-test or paired *t*-test (*p* < 0.05), whichever was appropriate.

## 3. Results

### 3.1. Serological Findings

The study utilized two different serum panels, i.e., A (n = 68) and B (n = 50), obtained from the suspected measles cases. Interestingly, Panel A had a documented history of measles immunization (at least one dose), whereas Panel B was unvaccinated. These serum panels were initially subjected to MeV-specific IgM and IgG antibody detection, subsequently challenged with measles vaccine (n = 1) and wild-types (n = 4) virus in neutralization experiments to establish homologous and heterologous Nt-Ab pattern (Table 1 and Table 2).

Panel A: In this panel, MeV specific IgM antibodies were detected in 17 subjects (out of 68) with fever and skin rash onset between 2 and 10 days (n = 10) and onset between 13 and 25 days (n = 7). All 17 subjects showed >1:128 Nt-Ab titres for MeV D4, 16 subjects showed >1:128 Nt-Ab titres for MeV D8 and 14 subjects showed >1:128 Nt-Ab titre for MeV vaccine virus. Interestingly, all these cases showed measles specific IgG and low IgG antibody avidity and are suggestive of recent MeV infection.

Of the remaining 51 MeV IgM negative cases, 21.5% of cases showed Nt-Ab under protective cutoff, whereas 78.4% cases showed above protective cutoff for MeV D8. However, 7.8% and 92.2% cases showed below and above protective Nt-Ab titres for MeV D4, respectively. Amongst these 51 cases, 39.2% and 60.8% cases showed below and above protective Nt-Ab titres for the MeV A (EZ) strain, respectively, whereas, 60.8% cases showed protective Nt-Ab titre for all three viruses. All 51 cases showed MeV-specific IgG with low, high and equivocal IgG-avidity in 7, 37 and 7 cases, respectively. Despite the absence of MeV-specific IgM antibodies in 75% cases (from Panel A), the presence of Nt-Abs indicates exposure to the virus.

Panel B: In this panel, MeV-specific IgM antibodies were detected in 26 subjects with fever and skin rash onset between 2 and 10 days (n = 13) and onset between 12 and 24 days (n = 13). Altogether, 25 and 26 subjects showed >1:128 Nt-Ab titres for the MeV D8 Jamnagar and New Delhi strains, respectively, whereas, 26 and 17 subjects showed >1:128 Nt-Ab titres for the MeV D4 Sindhudurg and Bagalkot strains, respectively. Of these cases (n = 26), 21 subjects showed IgG antibodies and low IgG-avidity.

The MeV-specific IgM antibodies were not observed in the remaining 24 fever with skin rash cases, of which 10 cases showed <1:128 Nt-Ab titre for the D8 Jamnagar and 9 cases for the New Delhi strains. However, 9 cases showed <1:128 Nt-Ab titre for the MeV D4 Sindhudurg and 10 cases for the MeV Bagalkot strains. Interestingly, 12 cases showed >1:128 Nt-Ab titre for the MeV D8 Jamnagar, 13 cases for the New Delhi, 15 cases for the Sindhudurg and 7 cases for the Bagalkot strains. Overall, 7 cases showed protective Nt-Ab titre for all four viruses. Interestingly, 18 of 24 cases showed IgG antibodies, however, there was low IgG-avidity in 4, high IgG-avidity in 10 and equivocal IgG-avidity in 4 cases. The absence of MeV-specific IgM antibodies in 48% cases (of panel B) indicates natural exposure due to the presence of Nt-Ab titre.

From this panel, two cases do not have Nt-Abs for both MeV D8 viruses (Jamnagar and New Delhi isolates) and also do not show IgM and IgG antibodies. Interestingly, a 4-year-old case (with 4 day’s onset) was low reactive, and another 6-month-old case (with 10 day’s onset) was high reactive in IgG antibody-avidity ELISA. These cases showed Nt-Abs for MeV D4 Sindhudurg but not for MeV D4 Bagalkot virus, clearly indicating natural exposure.

In contrast, seven cases showed Nt-Abs for the MeV D4 Sindhudurg but not for the MeV D4 Bagalkot virus. All these cases were IgM negatives but two of the seven were IgG positives. Of these, four cases were <1 years old and the remaining three cases were 2.1, 4.0 and 4.5 years old. Of these seven cases, two, three and two cases showed low, high and equivocal measles IgG antibody-avidity reactivity, respectively. Five of the seven cases showed Nt-Abs for both the MeV D8 viruses.

Overall, the GMTs of Nt-Ab titres for the MeV EZ vaccine strain were found to be significantly lower than MeV wild-types D4 and D8 viruses for panel A serum samples (Appendix A, paired *t*-test, *p* < 0.05 for each comparison). No major difference was noted in the Nt-Ab titres (GMTs, 198.8 vs. 201.3) between the MeV D8 Jamnagar and the New Delhi viruses (paired *t*-test, *p* > 0.05); in contrast, the MeV D4 Sindhudurg and Bagalkot viruses showed significant difference between Nt-Ab titres (GMTs, 288.2 vs. 80.6) for Panel B serum samples (Appendix A, paired *t*-test, *p* < 0.05). Thus, neutralization studies provide more information on the suspected measles cases.

### 3.2. Findings in the Laboratory Confirmed and Non-Confirmed Cases

In the laboratory confirmed (43 of 118) measles cases, 88.4% showed IgG antibodies and all showed Nt-Abs for both D4 and D8 viruses. Interestingly, 90.7% (39/43) showed low, 4.7% showed high (2/43) and 4.7% showed equivocal (2/43) IgG antibody avidity towards MeV.

Amongst 75 of the 118 not confirmed (IgM negatives) but clinically suspected measles cases, 92% showed IgG antibodies, whereas all the cases showed Nt-Ab for the wild-type D4 and 97.3% (73/75) cases showed Nt-Ab for the D8 viruses. Interestingly, 17.3% (13/75), 66.7% (50/75) and 16% (12/75) of cases, respectively, showed low, high and equivocal IgG antibody avidity towards MeV. This clearly indicates that IgM detection alone failed to measure measles reinfection or breakthrough cases and hence there is a need to utilize additional serological (avidity or NT) and molecular tools for case confirmation.

### 3.3. Variations in Known Epitopes of Indian MeV Isolates

The substitutions observed in H-protein sequences of four MeV isolates used in FRNT were analyzed in the context of known B cell epitopes [15,28]. As can be seen from Table 3, two isolates each belonging to D4 and D8 genotypes, L249P and G316A substitutions, are unique to the MeV Bagalkot (D4) isolate. This substitution is also present in seven Indian D4 isolates isolated during 2006–2007 (GenBank: FJ765087, FJ387132, FJ387137–FJ387140 and FJ387145). The L249P substitution is typical of the D4.2 sub-genotype and is observed in several global D4 isolates [29]. Older D4 isolates from 2003 (GenBank: AY594288 and KC291539) contain L249P and the extinct MeV genotype D6 also carried this substitution.

Seven substitutions were observed in experimentally validated B cell epitopes of fusion protein with respect to the Bagalkot, Sindhudurg and New Delhi isolates (Appendix A). Unique substitution S8Y was found in the Bagalkot isolate which is part of two known experimentally validated B cell epitopes [30,31,32].

### 3.4. Homology Modeling Results

The model of hemagglutinin protein for the MeV Bagalkot isolate was built using crystal structures of hemagglutinin glycoprotein Edmonston B strain of the measles virus (PDB ID: 2ZB6 [17] and PDB ID: 3ALZ [18]). The query sequence shared 65.1% identity and 65.4% similarity with 2ZB6 and an identity of 63% and 64.6% similarity with 3ALZ. The missing regions in the template structure 2ZB6 were 167–183 and 240–246 apart from the N-terminal and C-terminal regions. These missing regions were modeled as loops using 3ALZ. It is important to note that the NE and LE comprises residues 235, 244–250 and 309–318, respectively.

The second template 3ALZ was used to build missing residues in the query as well as in the template 2ZB6 prior to MD simulations. The predicted structures were checked for essential accuracy with respect to geometry and stereochemistry. The Ramachandran plot shows 89.5% residues in the allowed region and 0.5% residues in the disallowed regions, of which only GLN311 is part of LE. The ProSA Z-score of −7.9 and the total potential energy graph indicate that the model is optimized except for a few regions.

### 3.5. MD Simulation

The MD simulation of H-proteins for the MeV Bagalkot isolate and the Edmonston B strain was carried out in triplicate, indicating that both the NE and LE regions were found to adopt loop conformations. The RMSD analysis of the H-protein of Bagalkot and the Edmonston B strain showed significant difference where the latter was found to be more dynamic than the Bagalkot isolate (Appendix A). To study the effect of mutation in the NE (235, 244–250) and LE (309–318) regions between the Bagalkot isolate and the reference isolate, local RMSD analysis revealed that the NE regions showed significant fluctuations in both the isolates when compared to LE. As can be seen from (Figure 2a,b), the Edmonston B strain was more dynamic than the Bagalkot isolate, which harbors L249P substitution, as expected based on the side chain geometry. No significant differences in the RMSD were observed for the LE region containing substitution G316A, again as expected due to similar smaller side chains (Figure 3a,b). The per-residue fluctuations (RMSF) were calculated for the NE and LE, which showed that the L249P results in decreased conformational flexibility (RMSF range: 0.18–0.22) compared to L249 (RMSF range: 0.17–0.36). No significant conformational change was observed at G316A (RMSF range: 0.15–0.17) compared to G316 (RMSF range: 0.19–0.25).

## 4. Discussion

Complete genome sequencing of 58 MeV isolates from India revealed mutations in H-protein at important epitope sites, i.e., HNE, SSE, LE and NE for a few isolates [15,33]. This prompted us to undertake in-depth bio/immuno-informatics and neutralization studies on Indian MeV isolates. It is interesting to note that we have selected serum samples from the suspected measles cases (n = 118) that were referred for laboratory diagnosis. Subsequently, MeV-specific IgM, IgG, IgG-avidity and neutralizing antibody appearance were studied amongst ‘laboratory confirmed (IgM positives)’ and ‘non-confirmed (IgM negatives)’ cases. This is the first study which focused on detailed serological investigations on suspected measles cases and supplemented them with the bioinformatics of wild-type MeV isolates.

Previous reports have proposed that MeV specific Nt-Ab titre >1:120, measured using a plaque reduction neutralization test provides protection against clinical measles [34,35,36,37]. By considering this cutoff (here considered > 1:128), 82.7% and 70.7% of subjects showed protective Nt-Ab levels against the MeV wild-type (D8) and EZ (A) vaccine strains, respectively, for Panel A. Overall, 70.7% of subjects showed protective titres for both challenge viruses. Interestingly, all these subjects had clinical measles (i.e., fever with skin rashes) and serum samples were collected between 2 and 35 days of disease onset. It is important to note that only a single serum sample was collected from each of these cases for the serological testing, and subjects may have been either acute or convalescent.

The study noted marked differences in Nt-Ab titres amongst D4 (Sindhudurg vs. Bagalkot) and D8 (Jamnagar vs. New Delhi) wild-type viruses amongst Panel B; this may be due to changes noted in the F and H epitopes. However, it is interesting to note that the MeV Sindhudurg (Maharashtra) isolate was detected during the year 2012 and the MeV Bagalkot (Karnataka) isolate was detected during the year 2007. Interestingly, the panel of serum samples utilized in the neutralization experiments was from the state of Maharashtra, where circulation of the MeV genotypes D4 and D8 viruses has been reported [6,38,39]. Interestingly, not much difference in Nt-Ab titres was noted in Panel A between the two wild-type viruses (D4 and D8), however, Nt-Ab titres were about 50% lower for the MeV vaccine strain than for the wild-types, indicating either decay of immunity or likely non-responders amongst these one dose vaccine recipients. Similar observations in the neutralization pattern of different MeV vaccine and wild-type strains have been reported in Iran, the UK, Japan and the USA (11–14). However, for the first time, cases where measles was suspected were subjected to detection of the MeV-specific IgM, IgG, IgG-avidity and Nt antibodies (vaccine/wild-types) and their suitability for helping to achieve the measles elimination goal in India was described.

The observed substitutions in hemagglutinin and fusion protein that are part of experimentally validated B cell epitopes seem to be responsible for reduced titres for the MeV Bagalkot strain. This observation is further strengthened by the role of multiple co-dominant epitopes imposing constraints on MeV escape [40,41]. MD simulations provide an insight and explanation for lower Nt-Ab titres for the MeV Bagalkot strain wherein antigen–antibody induced fit movement increases the rigidity of epitope–paratope interactions. L249P substitution seems to reduce the conformational flexibility that differentiates MeV Bagalkot from the Sindhudurg strain. Further immuno-informatics (in silico) studies on antigen (wild-type MeVs)–antibody (IgG-subtypes) interactions may be explored. Interestingly, both T and B cell epitope immunoinformatics need to be studied to understand the antigenic pattern of circulating wild-type MeV isolates in the context of measles elimination.

Limitations of study: Considering the serologically monotypic nature of measles, both serum panels were not tested in neutralization experiments for all MeV viruses (i.e., Panel A was screened with one vaccine and two wild-types, whereas Panel B was screened with four wild-types based on the bioinformatics findings) and this remains one of the limitation of this study. Serum samples with large volumes that were referred for laboratory diagnosis of suspected measles cases were included in the study. Thus, the sample size estimation aspect was not considered and subsequently sera with history of measles vaccination (vaccinated and unvaccinated) were utilized in various serological assays to understand qualitative and quantitative outputs. In addition, other clinical specimens, i.e., urine and throat swabs were not available for MeV genotyping of these 118 cases. Due to the unavailability of 3rd World Health Organization international reference sera, MeV neutralizing antibody titres were not expressed in 120 mIU/mL and were expressed as 50% neutralizing antibody titres obtained using Kärber’s formula. Also, our study was mainly focused on cross-neutralization activity of various MeVs (wild-types and vaccine strains) and not on waning immunity phenomenon.

## 5. Conclusions

Neutralization activity for the MeV genotypes A (EZ vaccine), D4 (Sindhudurg-2012 and Bagalkot-2012) and D8 (Jamnagar-2016 and New Delhi-2014) viruses indicated good cross-neutralization in sera collected from clinically suspected measles cases. However, significant differences in the neutralizing antibody titres were noted within the MeV genotypes, i.e., D4 and D8 where mutations in H-protein (important epitopes, i.e., HNE, SSE, LE and NE) were evident. Molecular dynamics simulations revealed that the reduced neutralization can be attributed to the decreased conformational flexibility in the region belonging to neutralizing epitopes due to L49P substitution. Overall, wild-type MeVs are serologically monotypic, nevertheless, MeV whole genome sequencing and bioinformatics studies are crucial in the context of the measles elimination goal.

## Figures and Tables

**Figure 1 viruses-15-02243-f001:**
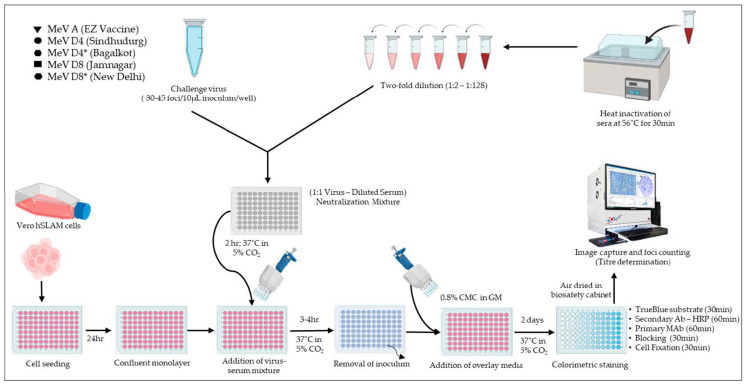
Schematic representation of measles FRNTs using five challenge viruses.

**Figure 2 viruses-15-02243-f002:**
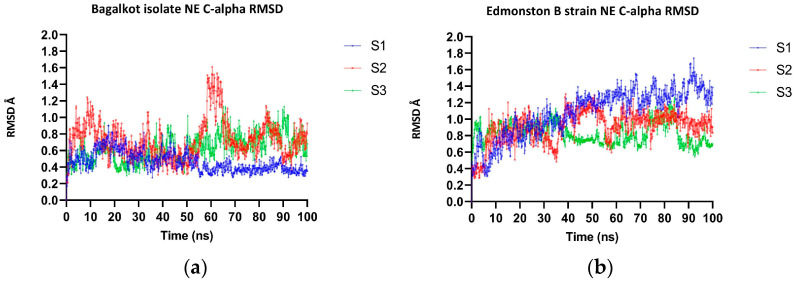
The figures contain RMSD plots of C-alpha of NE regions of MeV. (**a**) RMSD plot of C-alpha of NE region of MeV Bagalkot isolate H-protein. (**b**) RMSD plot of C-alpha of NE region of MeV Edmonston B vaccine strain H-protein.

**Figure 3 viruses-15-02243-f003:**
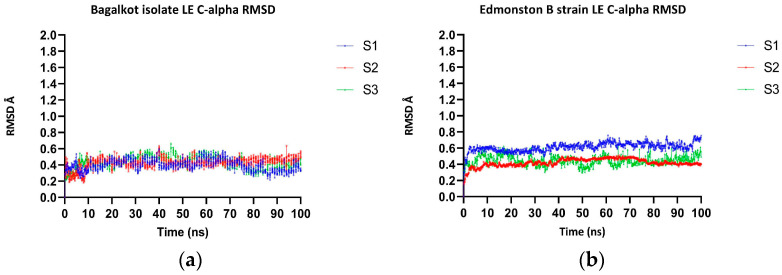
The figures contain RMSD plots of C-alpha of LE regions of MeV. (**a**) RMSD plot of C-alpha of LE region of MeV Bagalkot isolate H-protein. (**b**) RMSD plot of C-alpha of LE region of MeV Edmonston B vaccine strain H-protein.

**Table 1 viruses-15-02243-t001:** Age group based mean neutralizing titre for MeV challenge viruses (vaccine/wild-types) in Panel A and Panel B.

**Panel A**						
**Age Group** **(No. of Cases)**	**Mean Log_10_** **Titre ± SD (GMT)** **MeV A (EZ Vaccine)**		**Mean Log_10_** **Titre ± SD (GMT)** **MeV D4**		**Mean Log_10_** **Titre ± SD (GMT)** **MeV D8**	
<1 (6)	2.173 (148.85)		2.544 (349.55)		2.411 (257.71)	
1–6 (36)	2.199 (158.04)		2.546 (351.42)		2.430 (269.26)	
7–27 (26)	2.203 (159.42)		2.581 (380.95)		2.463 (290.23)	
Total/Overall (68)	2.198 (157.73)		2.559 (362.26)		2.441 (276.02)	
**Panel B**						
**Age Group** **(No. of Cases)**	**Mean** **Log_10_** **Titre ± SD (GMT)** **MeV D4**	**Mean** **Log_10_** **Titre ± SD (GMT)** **MeV D4 ***		**Mean** **Log_10_** **Titre ± SD (GMT)** **MeV D8**		**Mean** **Log_10_** **Titre ± SD (GMT)** **MeV D8 ***
<1 (15)	2.258 (181.15)	1.644 (44.03)		1.983 (96.08)		2.109 (128.54)
1–6 (35)	2.546 (351.75)	2.019 (104.52)		2.434 (271.52)		2.387 (244.02)
Total/Overall (50)	2.460 (288.25)	1.907 (80.64)		2.298 (198.81)		2.304 (201.33)

* With some important mutations.

**Table 2 viruses-15-02243-t002:** Post-onset days based mean neutralizing titre for MeV challenge viruses (vaccine/wild-types) in Panel A and Panel B.

**Panel A**						
**POD** **(No. of Cases)**		**Mean Log_10_** **Titre ± SD (GMT)** **A**		**Mean Log_10_** **Titre ± SD (GMT)** **D4**		**Mean Log_10_** **Titre ± SD (GMT)** **D8**
01–07 (19)		2.266 (184.65)		2.566 (367.89)		2.506 (320.97)
08–14 (18)		2.171 (148.10)		2.562 (364.68)		2.470 (295.45)
15–21 (14)		2.207 (161.00)		2.551 (355.67)		2.450 (281.60)
22–37 (10)		2.274 (187.82)		2.644 (440.45)		2.450 (281.52)
Total/Overall (61) $		2.226 (168.12)		2.574 (375.01)		2.473 (297.49)
**Panel B**						
**POD** **(No. of Cases)**	**Mean****Log_10_** **Titre** **± SD (GMT)** **MeV D4**		**Mean****Log_10_** **Titre** **± SD (GMT)** **MeV D4 ***		**Mean****Log_10_** **Titre** **± SD (GMT)** **MeV D8**	**Mean****Log_10_** **Titre** **± SD (GMT)** **MeV D8 ***
01–07 (13)	2.485 (305.21)		1.708 (51.07)		2.142 (138.75)	2.318 (208.18)
08–14 (19)	2.376 (237.70)		1.821 (66.27)		2.293 (196.13)	2.183 (152.25)
15–21 (10)	2.387 (244.01)		1.873 (74.72)		2.235 (171.96)	2.293 (196.51)
22–37 (8)	2.709 (511.44)		2.473 (297.03)		2.645 (441.64)	2.582 (381.67)
Total/Overall (50)	2.460 (288.25)		1.907 (80.64)		2.298 (198.81)	2.304 (201.33)

$ POD unknown for 7 suspected measles cases in Panel A. * With some important mutations.

**Table 3 viruses-15-02243-t003:** Variations observed in the B cell epitopes belonging to hemagglutinin protein of MeV isolates utilized in neutralization assays.

Epitopes in H-Protein	MeV Isolate			
	D8 (Jamnagar)	D8 * (New Delhi)	D4 (Sindhudurg)	D4 * (Bagalkot)
Hemagglutinating and noose	-	A400I	A400T	-
Sugar-shielded	-	I473V	I473V	I473V
Loop	Q315K	Q315K	Q315K	Q315K, G316A
Neutralizing	S247P	-	-	L249P

* With some important mutations.

## Data Availability

Original data are available upon reasonable request to the corresponding author.

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
