# Peer review of "Neutralizing Antibody Response to Genotypically Diverse Measles Viruses in Clinically Suspected Measles Cases"

_viruses, 2023, doi:10.3390/v15112243_

Round 1

Reviewer 1 Report

Comments and Suggestions for Authors

The authors have used serum samples of suspected measles cases, either or not with prior documented vaccination, to test neutralizing antibody levels to vaccine or wild-type measles virus strains belonging to genotypes D4 or D8. The results were interpreted in the context of homology modeling and molecular dynamics simulation of the H and F proteins of the different viruses. I have a few major and minor comments to the manuscript.

1. The authors describe an interesting set of experiments, but do not explain which question they are addressing. It would be useful to define a hypothesis and explain how the experiments were used to test this hypothesis. The question as explained in the last paragraph of the introduction is unclear to me: it mentions what was done but not why this was done, and why these two serum cohorts were selected for this study.

2. The question / hypothesis referred to in comment 1 should be used to streamline the introduction. The information in the current introduction is not incorrect, but lacks a “storyline”. The authors correctly point out that the virus is monotypic (line 306), and live-attenuated measles vaccines developed in the 1960’s still provide protection against currently circulating wild-type strains.

3. Paragraph 2.1: the authors should point out in this paragraph that each cohort contained both laboratory-confirmed measles cases and cases in whom acute MeV infection was not confirmed (potential “other rash disease” cases). Were any of the subjects also screened by Rt-PCR? The authors should add information on the time interval between onset of rash and collection of serum (now listed in line 293, but hopefully the authors can provide more detailed information). Was the time of sampling related to the outcomes of the experiments? Moreover, the authors should provide information on the predominant wt-MeV genotypes circulating in the region during the period of the study.

4. Paragraph 2.3: the authors should be aware of the existence of international reference sera for MeV neutralization. Presenting the outcomes as titers instead of international units per ml strongly diminishes the value of these studies. The authors should justify why reference sera were not used in this project. Protective cut-offs are exclusively defined on the basis of PRNT titers to the vaccine virus, not on basis of an in-house VNT against WT-MeV. Moreover, the cut-off was defined on antibody levels above 120 mIU/ml, only the first publication was based on titers (line 287-288).

5. Line 152: the absence of documentation of previous vaccination should not be confused for evidence that subjects in panel B were unvaccinated.

6. Line 161: I do not agree with the statement that detection of low avidity IgG antibodies is evidence for acute MeV infection. The only serological evidence for MeV infection is detection of IgM in a single sample or seroconversion / 4-fold titer rise in paired serum samples collected during the acute and convalescent phase. Low IgG avidity is at best suggestive for recent MeV infection.

7. The authors should consider displaying the results that are now listed in Tables 1 and 2 as dot plots, showing the VN titers for each individual sample of A vs D4, A vs D8 and D4 vs D8 (or D4 vs D4* and D8 vs D8*). Both titers should be plotted on a logarithmic axis. This would also show if the negatives were always negative against all viruses. Different symbols shapes or fills could be used to discriminate between subcategories (IgM = / -, age).

8. Paragraph 3.2: the authors should explain how these differences link back to their question or hypothesis (see my comment 1). Why are these subgroups analyzed separately and what does it tell us? The possibility of breakthrough infections has been described many times, so that message does not contain much novelty.

9. Discussion: the interpretation of the VN data in lines 289-291 should take into account how much time had passed between onset of rash and collection of the serum sample (see point 1). Were the antibodies in panel A induced by previous vaccination or by the recent wt-MeV infection? The range listed in line 293 is extremely wide: on day 2 the response to recent infection could still be absent, whereas at 35 days this would be expected to be really high.

10. The same holds true for the discussion in lines 302-305. I think the suggestion of decay of immunity or non-responders to vaccination is highly unlikely in this panel of samples from suspected measles cases, except in those cases where the serum was collected very early after onset of rash.

11. Discussion: many other studies have previously tried to detect differential VN antibody levels to different MeV genotypes, in most cases with limited success. In my opinion the common consensus is that these differences are either very small or non-existing. The authors should briefly discuss a number of these studies, and place their own data in the context of these findings.

12. The authors should try to use paragraph 5 (conclusions) to explain what their study adds to the international literature. The current paragraph is a listing of results without a global interpretation.

Comments on the Quality of English Language

English grammar can be improved.

Author Response

Attachment please.

Reviewer 2 Report

Comments and Suggestions for Authors

I was invited to revise the paper entitled "Neutralizing antibody response to genotypically diverse measles viruses in clinically suspected measles cases". It aimed ot evaluate  the in-vitro neutralization activity to 65 MeV vaccine (genotype A) and wild type (genotype D4). 

- Sample size estimation was totally lacking;

- Statistical analysis section needs improvements;

- Table 1 and 2 are totally ureadable! Authors should better describe what is presented in the table and wich variables were reported. 

- DIscussions section was poor. Authors should compare also their results with other study conducted in other countries.

Comments on the Quality of English Language

-

Author Response

Attachment please.

Reviewer 3 Report

Comments and Suggestions for Authors

General Comments:

This is an interesting study given that vaccine failures are increasingly documented together with the concern that the measles vaccines worldwide are derived from genotype A, that is no longer circulating.  As well the current circulating genotypes may be evolving and potentially not be adequately neutralized by the vaccine.

Unfortunately, the serum panels in this study are not adequately characterized and the overall result data is presented confusingly.

For example, Panel A apparently comprises of immunized individuals (one or two doses). However, the authors record that 17 of 68 were IgM positive with apparent clinical symptoms of measles which could be due to acquiring measles soon after vaccination when protective antibodies are still developing ?

As well it would have been very useful to have the MeV genotypes determined from a select number of cases in both Panels A and B to use as comparators when performing the neutralization studies with unknown status in this panel B and especially for the re-infected cases in Panel A.

I suggest that the authors review their serum panels and select or re-categorize their samples that are unambiguously a either lab-confirmed case or a vaccine failure and reanalyse the data accordingly.

Comments on the Quality of English Language

The quality of English was adequate and comprehensible

Author Response

Attachment please.

Reviewer 4 Report

Comments and Suggestions for Authors

The authors demonstrated a research article regarding neutralizing antibody titers in patients samples from India. The research question is important and the hypotheses are methodologically well addressed. However, there are some issues which need to be addressed:

1- the introduction section must be completely rewritten, most of the important measle virus papers are not cited and the biological structure of the virus and the deep clinical overview are missing.

2- a patient characterization should be added to the research article and the methods used for performing the neutralization assay should be explained in more detail and covered by a figure.

3- the limitation section is missing and the findings should be discussed in a more focused way.

Comments on the Quality of English Language

The English language is fine.

Author Response

Attachment please.

Round 2

Reviewer 1 Report

Comments and Suggestions for Authors

The authors have addressed most of the comments of the reviewers. 

Comments on the Quality of English Language

English language quality is improved.

Reviewer 2 Report

Comments and Suggestions for Authors

Authors addressed my previous comments. it can be accepted

Reviewer 4 Report

Comments and Suggestions for Authors

The authors have replied appropriately to my comments.